# Emerging Role of Vitamins D and K in Modulating Uremic Vascular Calcification: The Aspect of Passive Calcification

**DOI:** 10.3390/nu11010152

**Published:** 2019-01-12

**Authors:** Yi-Chou Hou, Chien-Lin Lu, Cai-Mei Zheng, Ruei-Ming Chen, Yuh-Feng Lin, Wen-Chih Liu, Tzung-Hai Yen, Remy Chen, Kuo-Cheng Lu

**Affiliations:** 1Department of Internal Medicine, Cardinal Tien Hospital, School of Medicine, Fu-Jen Catholic University, New Taipei City 23148, Taiwan; athletics910@gmail.com; 2College of Medicine, Fu-Jen Catholic University, New Taipei City 24205, Taiwan; janlin0123@gmail.com; 3Graduate Institute of Clinical Medicine, College of Medicine, Taipei Medical University, Taipei 11031, Taiwan; 11044@s.tmu.edu.tw (C.-M.Z.); linyf@shh.org.tw (Y.-F.L.); wayneliu55@gmail.com (W.-C.L.); 4Division of Nephrology, Department of Medicine, Fu Jen Catholic University Hospital, School of Medicine, Fu Jen Catholic University, New Taipei City 24205, Taiwan; 5Division of Nephrology, Department of Internal Medicine, Shuang Ho Hospital, Taipei Medical University, New Taipei City 23561, Taiwan; 6Graduate Institute of Medical Sciences, College of Medicine, Research Center of Cancer Translational Medicine, Taipei Medical University, Taipei 11031, Taiwan; rmchen@tmu.edu.tw; 7Anesthesiology and Health Policy Research Center, Taipei Medical University Hospital, Taipei 11031, Taiwan; 8Brain Disease Research Center, Wan-Fang Hospital, Taipei Medical University, Taipei 11031, Taiwan; 9Division of Nephrology, Department of Internal Medicine, Tri-Service General Hospital, Taipei 11490, Taiwan; 10Division of Nephrology, Department of Internal Medicine, Tungs’ Taichung MetroHarbor Hospital, Taichung City 435, Taiwan; 11Department of Nephrology, Chang Gung Memorial Hospital and College of Medicine, Chang Gung University, Taoyuan City 33305, Taiwan; m19570@adm.cgmh.org.tw; 12Kidney Research Center, Chang Gung Memorial Hospital, Taoyuan City 33305, Taiwan; 13Center for Tissue Engineering, Chang Gung Memorial Hospital, Taoyuan City 33305, Taiwan; 14Kidney Dialysis Center, Kamifukuoka General Hospital, Saitama 356, Japan; remyneko@yahoo.co.jp

**Keywords:** vascular calcification, vitamin K, calciprotein particle, matrix vesicle, vitamin D

## Abstract

Vascular calcification is a critical complication in patients with chronic kidney disease (CKD) because it is predictive of cardiovascular events and mortality. In addition to the traditional mechanisms associated with endothelial dysfunction and the osteoblastic transformation of vascular smooth muscle cells (VSMCs), the regulation of calcification inhibitors, such as calciprotein particles (CPPs) and matrix vesicles plays a vital role in uremic vascular calcification in CKD patients because of the high prevalence of vitamin K deficiency. Vitamin K governs the gamma-carboxylation of matrix Gla protein (MGP) for inhibiting vascular calcification, and the vitamin D binding protein receptor is related to vitamin K gene expression. For patients with chronic kidney disease, adequate use of vitamin D supplements may play a role in vascular calcification through modulation of the calciprotein particles and matrix vesicles (MVs).

## 1. Introduction

Chronic kidney disease (CKD) is defined as a progressive decline in the glomerular filtration rate. Patients with CKD have a higher mortality rate, and cardiovascular disease is the major cause of death among these patients [1]. Vascular calcification is a critical complication in patients with CKD. Calcification of arterioles within the endothelial and medial layers induces arterial stiffness and occlusion, which may cause coronary artery disease and calciphylaxis in peripheral vessels [2,3,4]. Therefore, vascular calcification is predictive of poor prognoses and clinical outcomes in CKD patients, such as overall mortality and even poor arteriovenous graft maturation [5,6]. Etiologies of vascular calcification include the traditional (e.g., hypertension, diabetes mellitus (DM), old age, smoking, and dyslipidemia [7]) and nontraditional risk factors (e.g., hyperhomocysteinemia, and higher high sensitive C-reactive protein (hsCRP) for cardiovascular disease [8]. Vitamin D and K deficiencies also serve as risk factors for uremic vascular calcification [9,10]. Recently, the roles of calciprotein particles (CPPs) and matrix vesicles (MVs) in vascular calcification have been widely discussed [11,12]. Furthermore, vitamin K supplements were found to be crucial for stabilizing CPPs and MVs in patients with CKD [13]. Through this review, we elucidate the relationship between vitamin K deficiency and uremic calcification, as well as the role of vitamin supplements in the prevention of uremic vascular calcification.

### 1.1. The Traditional Aspect: Active Uremic Vascular Calcification Induced by Hyperphosphatemia and Uremic Toxin Accumulation

Ingested phosphate is absorbed from the gastrointestinal tract and transported to bone tissue to facilitate the bone remodeling process [14]. The excess phosphate is excreted through the kidney. As glomerular filtration decreases, the decreased clearance of phosphate from the kidneys activates phosphaturic hormones, including FGF23 and parathyroid hormone, to increase the renal clearance of phosphate [15]. However, in advanced CKD, activated parathyroid hormone accelerates bone resorption rather than bone formation. Thus, osteoblasts cannot complete their mineralization process [16]. The accumulated phosphate activates the osteogenic differentiation of vascular smooth muscle cells (VSMCs) directly or through the activation of the renin–angiotensin–aldosterone system (RAAS), which is related to the inflammation process [17]. Such osteogenic differentiation induces active calcification within the medial layer. Clinical evidence has suggested that higher dietary phosphate ingestion is related to more severe vascular calcification [18].

Beyond hyperphosphatemia, protein-bound uremic toxins accelerate uremic vascular calcification [19]. Protein-bound uremic toxins (PBUTs), such as indoxyl sulfate and p-cresol sulfate, are derived from amino acids (such as tryptophan and tyroxine/phenylalanine) metabolized by the intestinal flora [20]. Subsequently, the metabolites are transformed into PBUTs in the liver, and then excreted in urine by organic anionic transporters in the renal tubules. In patients with renal insufficiency, accumulated PBUTs will directly damage endothelial cells by (a) inducing reactive oxidative stress within the endothelial layer [21,22]; (b) producing microparticles that interfere with nitric oxide released from endothelial cells [23,24]; and (c) impairing endothelial progenitor cells released from the bone marrow [25]. The damaged endothelial cells induce the proliferation and migration of VSMCs from the medial layer, as well as impair the integrity of the endothelial layer. Such a process induces active calcification within the endothelial and medial layers of vessels. 

### 1.2. The New Aspect: Passive Calcification through Interaction between CPPs and MVs

In advanced CKD, accelerated bone resorption caused by secondary hyperparathyroidism increases the amount of calcium and phosphate released from bone tissues. Moreover, protein-binding uremic toxins abate the mineralization of osteoblasts and worsen extraosseous calcium-phosphate crystal deposition [26]. Such passive calcification, in contrast to active calcification caused by osteogenic transformation, involves CPPs and MVs [27].

Protein-binding uremic toxins and hyperphosphatemia initiate the active calcification process within vascular smooth muscle layers. To counteract the calcification process, several inhibitors underlie the serum, including secretory Klotho protein, matrix Gla protein (MGP), osteopontin, osteoprotegerin, fetuin-A, vitamin K, pyrophosphate, and magnesium [28]. A CPP, as a calciprotein monomer (modified after) and low-density lipoprotein particle, serves as the solid phase of the calcium–phosphate compound [11]. Calciprotein interacts with calcification inhibitors, such as fetuin-A and MGP, to integrate the CPPs into spherical rather than unstructured agglomerates of minerals (clusters) and fetuin-A/MGP [29,30]. As the Figure 1, such CPPs (primary CPPs) may be cleared by the liver through scavenger receptor A present on hepatic endothelial cells [11]. In subjects with dysregulated or insufficient fetuin-A/MGP, the primary CPPs would be transformed to a crystalline mineral core (secondary CPP), which is not excreted through the liver. CPP1 is round with a diameter of 60–75 nm and is mainly composed of amorphous calcium phosphate, which exists as a colloid. CPP2 includes hydroxyapatite calcium that results from calcium phosphate crystallization over time in the core and has a diameter of 120–150 nm. The strong negative charges in the β sheet of the cystatin-like D1 domain of fetuin-A are thought to promote binding with calcium phosphate and prevent its growth, aggregation, and precipitation. Compared with CPP1, CPP2 has a lower level of fetuin-A and surface charge, and a higher apolipoprotein content. Therefore, the higher proportion of CPP2 influences endothelial damage directly, as well as exacerbating further calcification by recruiting inflammatory cells, such as macrophages, through TLR-4 activation [31]. The accumulated secondary CPPs are deposited in the extracellular matrix, and exacerbate extraosseous calcification. Therefore, an insufficient level of calcification inhibitors within the serum are predictive of vascular calcification [32]. Yamada et al. provided evidence that diet restriction in CKD contributes to insufficient formation of calcification inhibitors, which is correlated with more severe vascular calcification [33]. Furthermore, Chen et al. provided evidence that the transformation of CPP1 to CPP2 was faster in patients with CKD, and such accelerated transformation was associated with vascular calcification. From the post-hoc analysis of the EVOLVE trial, the shorter time for the transformation of CPP1 to CPP2 was predictive of mortality due to uremic vascular calcification in patients with CKD receiving calcimimetics [34]. Recently, Ruderman et al. reported regarding the variation of primary and secondary CPPs, and the clinical event of vascular calcification in CKD patients. In patients who had discontinued calcimimetics for longer than 12 months, the serum levels of parathyroid hormone as well as CPP1 increased. In these patients, the 1-year mortality rate was 19%. Although no direct evidence was provided that mortality was directly related to the withdrawal of calcimimetics, the associated change in CPPs should be noted in patients with CKD as vascular calcification [35]. According to the abovementioned evidence, CPP transformation plays a crucial role in uremic vascular calcification, and further investigation of the effect of modulation of primary CPPs, rather than CPPs, is required. 

After secondary CPPs are deposited in the extracellular matrix, sequentially-released MVs play a role in modulating the VSMC calcification process [36].

MVs are a subgroup of extracellular vesicles (EVs) enclosed in a double membrane, and are composed of phosphatidylserine and annexin [37]. Various cells release vesicles into the extracellular environment to cope with cellular apoptosis [38]. Hematopoietic cells within the vessels, such as endothelial progenitor cells, monocytes, platelets, and red blood cells, release EVs. The released EVs interact with endothelial cells and induce endothelial dysfunction through oxidative stress, ICAM-1, or other chemokines. In the arterial medial layer, excess phosphates enter VSMCs through endocytosis, and the calcium released from lysosome activates osteogenic expression, and releases MVs into the extracellular matrix [39]. In patients with CKD, hyperphosphatemia and extraosseous calcium enter the VSMCs, and then induce intracellular oxidative and endothelial reticular (ER) stress [40]. Such oxidative stress increases the cellular release of calcium and phosphate into the extracellular matrix through MVs. In patients with CKD, MVs contain less fetuin-A or Gla-rich proteins (GRP), and such MVs were shown to be related to higher severity of mineral calcification in soft tissue [13]. In addition, the EVs in the serum of CKD patients were found to be prone to vascular calcification because they carried a higher percentage of calcification-associated markers, such as GRP. Because hyperphosphatemia and dysregulated calcium deposition directly activate vascular calcification, further interventions focusing on CPPs and MVs should be considered to prevent uremic calcification.

### 1.3. Vitamin D and K Deficiencies in CKD as an Etiology of Vascular Calcification

#### Metabolism of Vitamin K in CKD

Vitamin K is essential for the post-translational conversion of peptide-bound glutamate to γ-carboxyglutamate. Vitamin K is a lipophilic molecule consisting of the 2-methyl-1,4-naphthoquinone nucleus and a polyisoprenoid side chain at the 3-position. In nature, plant forms (vitamin K1 and phylloquinone) and bacterial forms (vitamin K2 and menaquinones) are the isoprenologs of vitamin K. After being ingested from the intestine, vitamin K is transported to the liver with triacylglycerol-rich lipoprotein, where vitamin K1 is transformed into vitamin K2 [41]. The transformed vitamin K2 is transported to extrahepatic tissues, such as bones, arteries, and macrophages, by low-density lipoproteins [42]. In extrahepatic tissue, menaquinone-4 is the major form of vitamin K with a mode of action. Vitamin K involves the carboxylation process with vitamin K-dependent protein (VKDPs). In patients with a vitamin K sufficient status, γ-glutamate is carboxylated to γ-carboxyglutamate, and then transported into the extracellular space [43]. After being carboxylated by vitamin K-dependent protein, phosphorylated MGP serves as a calcification inhibitor in several respects by (1) directly serving as a chelator for calcium and calcium crystals; (2) binding to bone morphogenetic protein-2 (BMP-2) to avoid osteoblastic differentiation of VSMCs; (3) exerting an antiapoptotic effect; and (4) generating CPPs with a lower percentage of secondary forms with the help of γ-carboxyglutamate, through CPPs/MVs [44] (Figure 2).

In vitamin K-deficient patients, the carboxylation process is hampered, and vascular calcification may be worsened. Insufficient levels of vitamin K are associated with more severe cardiovascular complications. In patients with DM, uncarboxylated MGP increases within the serum, which has been associated with arterial stiffness [45,46].

As previously mentioned, vitamin K1 is mainly found in green leafy vegetables, and vitamin K2 is mainly found in the fermented dairy such as cheese. However, the potassium concentration is higher in these foods [1]. Therefore, vitamin K deficiency is common in patients with CKD because they consume fewer vegetables due to the dietary potassium restriction [47]. Only a vegan or very low protein diet (<0.3 g protein/kg body weight) can provide high vitamin K1 content [48]. Moreover, vitamin K deficiency, along with other biomarkers of malnutrition, is common in CKD patients with anorexia or gastrointestinal dysfunction [49]. Subclinical vitamin K deficiency has been shown to be common in patients with CKD, hemodialysis (HD), and peritoneal dialysis (PD) [49,50,51], and furthermore, the vitamin K concentration in serum was not correlated with lipid profiles such as triglyceride or high-density lipoproteins (HDL) [50]. A clinical study showed that vitamin K1 concentration was significantly lower in hemodialysis patients [51]. Moreover, in patients with CKD, menaquinone concentration was not elevated after sustained ingestion of a diet rich in vitamin K for 7 weeks [52]. Based on the aforementioned evidence, functional and qualitative vitamin K deficiency is common in CKD patients. Simultaneously, vitamin K deficiency is predictive of vascular calcification in these patients. Nigwekar et al. provided evidence that the fraction of total MGP that is carboxylated is predictive of calciphylaxis in patients with end-stage renal disease (ESRD) [53]. In contrast, in animal studies, vitamin K metabolism is altered under CKD status. McCabe et al. reported the different distribution patterns of vitamin K isoforms in rats with CKD. Furthermore, the expression of vitamin K recycling (Vkor) and utilization (Ggcx) enzymes in the thoracic aorta of rats with CKD decreased [52]. Kaelser et al. demonstrated that γ-carboxylase activity reduced in the liver and kidney of adenine-treated rats. This decreased γ-carboxylase activity was associated with aortic calcification. In CKD patients, uncarboxylated MGP was associated with arterial stiffness, and after vitamin K supplementation, the uncarboxylated MGP levels decreased [54]. In renal transplant patients, vitamin K concentration may be lower than that in the normal population, and insufficient vitamin K levels were associated with higher dephospho-uncarboxylated MGP [55]. Although no direct evidence has demonstrated the uremic milieu’s influence on γ-carboxylase activity within tissues or γ-carboxylase activity expression in humans, vitamin K deficiency is common in CKD patients.

In clinical studies, vitamin K deficiency has been associated with vascular calcification. According to cross-sectional studies, the amount of ingested phylloquinone or menaquinones was predictive of coronary artery calcification [56,57]. Rattazzi et al. provided evidence that warfarin, a vitamin K antagonist, worsened aortic valve calcification in a mouse model of atherosclerosis [58]. Vitamin K supplements attenuated vascular calcification by suppressing Toll-like receptors in an atherosclerosis animal model [59]. These studies have provided evidence that vitamin K is functionally deficient. In CKD patients, vitamin K deficiency is associated with vascular calcification. Nigwekar et al. conducted a nationwide study on the correlation between vitamin K antagonists and the incidence of vascular calcification [60]. In CKD animal models, vitamin K antagonists influenced the vasculature. For example, Zaragatski et al. provided evidence that vitamin K antagonists worsened neointimal hyperplasia in rats with CKD [61]. Moreover, vitamin K-dependent carboxylation of osteocalcin (OC) modulated the bone remodeling status. In high turnover diseases, carboxylated osteocalcin promoted bone formation and mineralization. Thus, less calcium and phosphate would be released into the vasculature, and vascular calcification might be hampered. Vitamin K supplementation may reverse uncarboxylated MGP in CKD patients [62]. In 2012, a double-blind, multicenter controlled trial was initiated to validate the effect of vitamin K (10 mg of phylloquinone three times per week) for the prevention of coronary artery calciphylaxis in ESRD patients [2]. Nigwekar et al. also initiated a single-center clinical trial on a vitamin K supplement in uremic vascular calcification in ESRD patients (ClinicalTrials.gov identifier: NCT02278692). The estimated study completion dates of both studies are in 2019, and the results of these studies might answer if the vitamin K supplement could alleviate uremic vascular calcification.

### 1.4. Vitamin D Supplementation as a Potential Target for Salvaging Uncarboxylated MGP

Vitamin D deficiency is a common complication in CKD patients because of (1) proteinuria [63,64], (2) decline in the glomerular filtrate rate [65], (3) tubulointerstitial injury [66], and (4) therapeutic dosage of active vitamin D [67]. Vitamin D deficiency is associated with multiple complications in CKD patients, including infection, endothelial dysfunction, impaired myocardial remodeling, and insulin resistance. Vitamin D is defined based on the serum 25-hydroxyvitamin D (25(OH)D) concentration. Based on clinical evidence, vitamin D deficiency has been noted to be an etiology of vascular calcification [68]. Hypovitaminosis is associated with more advanced cardiovascular disease because of insulin resistance [69] and activation of the renin–angiotensin–aldosterone system [70]. Insulin resistance is associated with reduced endothelial response to shearing stress. Regarding the RAAS system, vitamin D deficiency is associated with its activation in DM nephropathy animal models and DM nephropathy patients. In pediatric patients with CKD, vitamin D deficiency was associated with higher incidence of arterial stiffness [71]. In that study’s model, confounding factors for uremic vascular calcification such as aging or insulin resistance were excluded. Thus, vitamin D deficiency should be a contributing factor for uremic vascular calcification.

Vitamin D has been demonstrated to interact with vitamin K-dependent proteins (Figure 3). Previous basic studies showed that the rate of ostelcalcin and MGP secretion increased after treatment with vitamin D. Fraser et al. first noticed that the protein and mRNA expression of MGP in osteosarcoma cell line increased after 1.25(OH)2D treatment at the concentration of 0.3 nM for more than 48 h. They also found vitamin D can stimulate osteocalcin synthesis earlier and at lower vitamin D levels compared to MGP production. [3]. In osteoblasts, 1.25(OH)2D increases vitamin K-dependent binding protein-related osteocalcin expression, which could maintain the mineralization of osteoblasts into osteocytes [72,73]. In a recent study, vitamin K helped osteogenesis of human mesenchymal stem cells by activating vitamin D3-mediated osteocalcin release [74]. Furthermore, Poon et al. showed that the conjunction of vitamin K(2) with 1.25(OH)2D increased osteoblast anabolism in diabetic rats [75]. By enhancing the bone anabolism, the extraosseous calcification might be lessened. On the other hand, a pharmacologic dosage of 1.25(OH)D supplement induced excessive intestinal absorption of calcium and phosphate. Such absorption is related to extraosseous calcification at the same time. Semaya et al. also provided the in vivo evidence that vitamin K supplement alleviated the experimental calcinosis induced by vitamin D with dosage of 2.5 × 10(5) I.U./kg b.w [4]. Besides, MGP directly inhibited the osteoblastic differentiation of the VSMCs [5]. Since vitamin D directly stimulates the vitamin K-dependent MGP production [6] in vivo and in vitro, supplying vitamin D could alleviate vascular calcification. 

Clinical evidence suggests that vitamin D deficiency has a synergistic effect on worsening clinical outcomes in vitamin K deficiency. Van Ballegooijen et al. provided evidence that vitamin D deficiency (<50 mmol/L), along with vitamin K deficiency, predicted higher blood pressure and higher risk of hypertension in the Netherlands [7]. O’Connor et al. provided evidence that 25(OH)D deficiency was associated with lower serum uncarboxylated osteocalcin and lower bone mineral content in Danish girls, although a vitamin D supplement did not increase serum osteocalcin (400 IU/day for 12 months) [76]. In a cross-sectional study, Mayer et al. noticed that insufficient 25(OH)D levels were associated with higher serum levels of dephospho-uncarboxylated matrix γ-carboxyglutamate protein, and thus were associated with higher aortic pulse wave velocity and aortic stiffness, and the polymorphism of vitamin D receptor (GG phenotype) along with vitamin K deficiency predicted higher aortic pulse wave velocity [77]. Because deficiencies of vitamins D and K play a conjunctive role in osteoporosis and vitamin K-dependent protein metabolism, vitamin D or K supplementation could alleviate vascular calcification in CKD patients. Asemi et al. demonstrated that synergic supplementation of vitamins D and K improved the insulin sensitivity and carotid intima-medial thickness in type 2 diabetic patients [78]. Furthermore, daily dosages of 5 µg of vitamin D and 90 µg of vitamin K2 for 12 weeks improved vascular thickness. In addition, a cross-sectional study of Italian hemodialysis patients demonstrated that treatment with vitamin D analogs (20%) was associated with higher percentage total and uncarboxylated osteocalcin concentrations, however, no significant association with total and uncarboxylated MGP was observed [79]. In CKD patients, CKD-MBD caused by high or low bone turnover disease impaired the normal bone remodeling process. In high bone turnover diseases, secondary hyperparathyroidism activates receptor activator of nuclear factor kappa-Β ligand (RANKL) signaling as well as osteoclast activity to increase bone resorption [80]. In low bone turnover diseases, inert osteoblast activity decreases the utility of calcium and phosphate, and subsequently decreases the extraosseous calcium-phosphate deposits [81]. Activating osteoblast activity is crucial for bone remodeling recovery. Gigantes et al. reported that co-supplementation of vitamins D and K is helpful for osteogenesis because vitamin K enhances the vitamin D gene induction of osteocalcin in mesenchymal stem cells. Moreover, vitamin K enhances osteogenesis and further mineralization [74]. Furthermore, in an ex vivo study, the synergic effects of vitamins D and K were able to alleviate the formation of advanced glycoxidated end products in osteoblasts. Thus, the decreased end product could improve bone health [82]. Vitamin D also alleviated secondary CPP formation in recipients of renal transplantation [83]. Based on the clinical trials above, vitamin D supplements improved the serum concentration of MGP and osteocalcin. Since MGP alleviates the vascular calcification directly and VSMC or osteocalcin maintained the bone health by decreasing extraosseous calcification, vitamin D supplements should be a promising target by increasing vitamin K-dependent calcification inhibition.

## 2. Conclusions

Uremic vascular calcification is a critical complication in patients with CKD and is predictive of multiple morbidities and a higher mortality rate. Beyond the traditional risk factors such as uremic toxin or hyperphosphatemia, the importance of CPPs and MVs has gradually grown. Vitamin K is vital for maintaining matrix gamma-carboxylation, and vitamin K deficiency is common in CKD patients both functionally and quantitatively. Vitamin K-dependent gamma-carboxylation has been noted to be modulated by vitamin D-binding protein-related gene expression. In CKD patients with vascular calcification, vitamin D supplementation may be a possible therapeutic target for restoring matrix gamma-carboxylation along with vitamin K to alleviate uremic vascular calcification.

## Figures and Tables

**Figure 1 nutrients-11-00152-f001:**
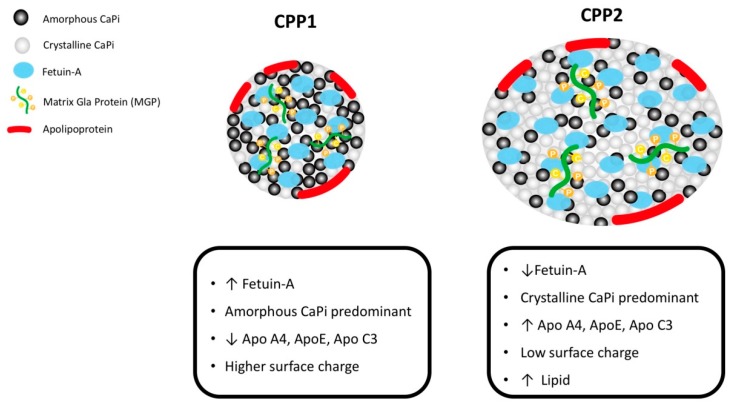
Components of calciprotein particles (CPPs): CPPs are composed of 18% mineral, 80% fetuin-A, and 2% matrix Gla protein [11]. Primary CPP (CPP1) is round with a diameter of 60–75 nm and is mainly composed of amorphous calcium phosphate, which exists as a colloid. Secondary CPP (CPP2) includes hydroxyapatite calcium that results from calcium phosphate crystallization over time in the core and has a diameter of 120–150 nm. The strong negative charges in the ***β*** sheet of the cystatin-like D1 domain of fetuin-A are thought to promote binding with calcium phosphate as well as prevent its growth, aggregation, and precipitation. Compared with CPP1, CPP2 has a lower level of fetuin-A, lower surface charge, and higher apolipoprotein content.

**Figure 2 nutrients-11-00152-f002:**
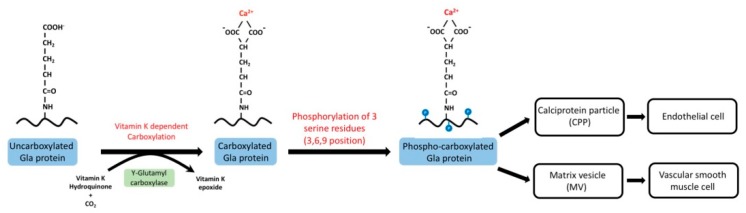
Phosphorylation and carboxylation process of matrix Gla protein (MGP) [43]. Vitamin K is a crucial cofactor for the post-translational γ-carboxylation of glutamic acid residues in MGP. Vascular calcification in the extracellular matrix is initiated by the deposition of minerals and MGP-containing calciprotein particles (CPPs) and matrix vesicles (MVs). Mineral nucleation within CPPs and MVs is blocked in the presence of mineral inhibitors, such as phosphorylated carboxylated MGP (p-c MGP) and fetuin-A [44]. However, uncarboxylated MGP and fetuin-A deficiency in chronic kidney disease (CKD) result in an increased level of mineral maturation followed by the calcification of vascular smooth muscle cells [45].

**Figure 3 nutrients-11-00152-f003:**
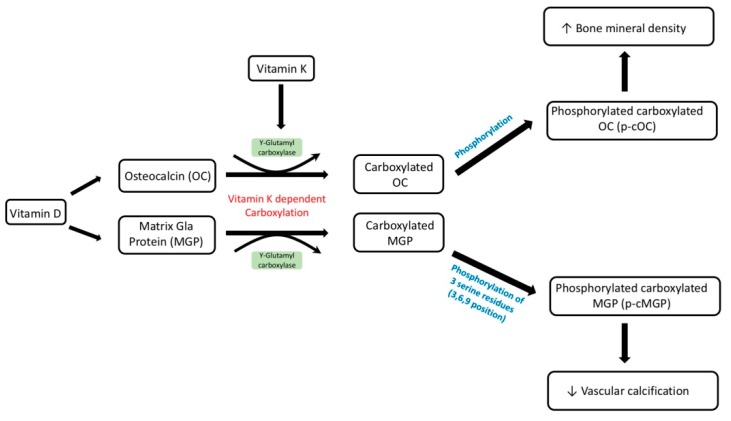
Synergistic effect of vitamins D and K on bone and vascular calcification. Vitamin D has been demonstrated to interact with vitamin K-dependent protein (VKDPs). VKDPs include several clotting factors and bone and soft tissue mineralization proteins, such as osteocalcin (OC) and matrix Gla protein (MGP) [74]. OC is carboxylated by vitamin K, and carboxylated OC is produced by osteoblasts. Carboxylated OC binds to hydroxyapatite in the extracellular matrix fob one. MGP is found mainly in normal vascular smooth muscular cells and upregulated during calcification [5]. Vitamin D supplementation is beneficial for OC and MGP carboxylation and is further phosphorylated to the phosphorylated carboxylated product (p-c OC and p-c MGP), which can further improve bone mineralization and alleviate vascular calcification in CKD patients.

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
