# Peer review of "Emerging Role of Vitamins D and K in Modulating Uremic Vascular Calcification: The Aspect of Passive Calcification"

_nutrients, 2019, doi:10.3390/nu11010152_

Round 1

Reviewer 1 Report

This review by Hou et al is a narrative review that describes novel aspects of vascular calcification and the potential role of both vitamin D and K on vascular calcification particularly in chronic kidney disease patients. This topic is interesting, however the mechanisms of how vitamin D acts upon vitamin K dependent proteins and the already published studies in the field should be described in more depth.

Abstract

Include that not many human studies have been performed that investigated both vitamin D and K on vascular outcomes and that only 1 RCT investigated the effect of combined supplementation on IMT.

Line 56: don’t use the word wish for the aim. Use more scientific words

Line 187: Only a vegan diet can provide high vitamin K1. Please also include vitamin K2 and important fermented dairy sources. Further vitamin K2 has a higher bioavailability. Curr Nutr Rep. 2017;6(3):197-205.

Line 205: vitamin K supplementation may be lower? Concentrations perhaps?

Lines 224-281 is the novel part of this review and should be described more in detail and in more depth. Many conclusions and misleading and study designs and vitamin D dose, and study duration should be mentioned.

Figures 3: lines 246-249, this conclusion cannot be drawn and should be toned town. Further, explain in more detail what does vitamin D do for vitamin K dependent proteins. Increase the synthesis, concentrations, gene expression. Give the reader a better sense of the mechanism. Use some mechanistic studies to explain this. Calcif Tissue Int 1990, 46, 270-279.7; Int J Vitam Nutr Res 1996, 66, 36-38.

Include some more human studies that investigated the joint associations of both vitamin D and K for vascular health: Hypertension. 2017 Jun;69(6):1165-1172.

Lines 257-259 about Danish girls. The conclusion is misunderstood and vitamin D has no effect on osteocalcin in this study. Among Danish girls  no effect of daily supplementation of 10 mg (400 IU) vitamin D3 vs. placebo on serum percentage undercarboxylated osteocalcin was found after 12-months. Please, rephrase.

Lines 259-261: the study should be described in more detail and is crossectional. Mention as such.

Lines: 267-269 again this conclusion is misleading: In a cross-sectional study among Italian hemodialysis patients, treatment with vitamin D analogs (20%) was associated with higher percentage total and uncarboxylated osteocalcin concentrations, however, no significant association with total and uncarboyxlated MGP was observed [15]. Rephrase!

Line 279-281: This conclusion is not based on the previous paragraph. Please, give an appropriate summary and conclude that vitamin D supplementation combined with vitamin K might be promising.

Reviewer 2 Report

In this review the Authors wish to elucidate the relationship between vitamin K deficiency and uremic calcification as well as the role of vitamin supplements in the prevention of uremic vascular calcification.  They conclude that in CKD patients, vitamin D supplementation may be a possible therapeutic target along with vitamin K to alleviate uremic vascular calcification. 

Along the extensive and detailed explanation of the role of vitamin D and K in CKD patients, the Authors should mention that currently there is a large multicenter RCT recruiting incident hemodialysis patients (Inhibiting the progression of arterial calcification with vitamin K in HemoDialysis patients [iPACKHD]) comparing 10 mg vitamin K1 treatment thrice weekly vs placebo, with the primary endpoint as change in CAC. Another clinical trial is also underway evaluating the effect of vitamin K1 compared to placebo on CUA (ClinicalTrials.gov identifier NCT02278692).

Minor point:

Figure titles are repeated twice.

Round 2

Reviewer 1 Report

Thank you for the revised manuscript. All my points have been addressed appropriately.